# Ammonium and Phosphonium Salts Containing Monoanionic Iron(II) Half-Sandwich Complexes [Fe(η$^5$-Cp*)X$_2$]$^-$ (X = Cl − I)

Julian Zinke, Clemens Bruhn and Ulrich Siemeling *

Institute of Chemistry, University of Kassel, Heinrich-Plett-Str. 40, 34132 Kassel, Germany;
julianzinke@uni-kassel.de (J.Z.); bruhn@uni-kassel.de (C.B.)
* Correspondence: siemeling@uni-kassel.de

**Abstract:** Half-sandwich iron(II) dihalido complexes of the type [Fe(η$^5$-Cp′)X$_2$]$^-$ (Cp′ = C$_5$H$_5$ or substituted cyclopentadienyl) which are thermally stable at room temperature are extremely scarce, being limited to congeners containing the bulky C$_5$H$_2$-1,2,4-*t*Bu$_3$ ligand. We extended this to homologues [Fe(η$^5$-Cp*)X$_2$]$^-$ (X = Cl, Br, I) containing the particularly popular C$_5$Me$_5$ (Cp*) ligand. Corresponding ionic compounds ER$_4$[Fe(η$^5$-Cp*)X$_2$] are easily accessible from FeX$_2$, MCp* (M = Li, K) and a suitable halide source R$_4$EX (E = N, P) in THF. Despite their high sensitivity towards air and moisture, the new compounds N*n*Pr$_4$[Fe(η$^5$-Cp*)X$_2$] (X = Cl, Br), N*n*Pr$_4$[Fe(η$^5$-Cp*)BrCl], and PPh$_4$[Fe(η$^5$-Cp*)X$_2$] (X = Cl, Br, I) were structurally characterised using single-crystal X-ray diffraction. N*n*Pr$_4$[Fe(η$^5$-Cp*)Cl$_2$] reacts readily with CO to afford [Fe(η$^5$-Cp*)Cl(CO)$_2$], indicating the synthetic potential of ER$_4$[Fe(η$^5$-Cp*)X$_2$] in FeCp* half-sandwich chemistry.

**Keywords:** crystal structures; cyclopentadienyl complexes; half-sandwich complexes; halides; iron

## 1. Introduction

Half-sandwich iron(II) complexes of the type [Fe(η$^5$-Cp′)X] (Cp′ = C$_5$H$_5$ or substituted cyclopentadienyl; X = Cl, Br, I) are useful as highly reactive cyclopentadienyliron(II) transfer reagents, which, due to their thermal lability, are usually generated in situ at low temperatures for immediate use [1]. Seminal work was published in 1985 by Kölle, who described the generation of [Fe(η$^5$-Cp*)Br] (Cp* = C$_5$Me$_5$) from LiCp* and [FeBr$_2$(DME)] in THF at −80 °C [2]. The corresponding chlorido complex [Fe(η$^5$-Cp*)Cl], which suffers from the same thermal lability, is particularly popular as a Cp*Fe$^+$ source [3–10]. The analogous complex containing the pentamethylcyclopentadienyl-related and *O*-donor-functionalised ligand C$_5$Me$_4$[(CH)$_2$(OCH$_2$CH$_2$)$_3$OMe], whose oligoether chain is suitable for intramolecular chelation, was reported to be thermally stable in THF solution up to room temperature, although this complex could not be isolated [11]. In contrast to the pronounced thermal lability of [Fe(η$^5$-Cp*)Cl], its *N,N,N′,N′*-tetramethylethylenediamine (TMEDA) chelate [Fe(η$^5$-Cp*)Cl(TMEDA)] is perfectly stable at room temperature [12], and the same holds true for the closely related complexes [Fe(η$^5$-C$_5$Me$_4$Et)Cl(TMEDA)] [13] and [Fe(η$^5$-Cp*)Br(TMEDA)] [14]. Note that corresponding *P,P*-coordinated complexes are generally more robust (but still not air-stable) and can be isolated even when containing an unsubstituted cyclopentadienyl (Cp) ligand, typical examples being the 1,2-bis(diphenylphosphanyl)ethane (DPPE) chelates [Fe(η$^5$-Cp)X(DPPE)] (X = Cl, Br, I), which were reported more than five decades ago [15,16]. Similar to the aforementioned TMEDA-containing *N,N*-chelates, *C,N*-chelates [Fe(η$^5$-Cp*)X(NHC$^N$)] (X = Cl, I) containing N-heterocyclic carbenes functionalised with an *N*-donor moiety (NHC$^N$) have also been described [17–19], and unchelated analogues [Fe(η$^5$-Cp*)Cl(NHC)] proved sufficiently stable for isolation with the standard NHC IMes and the bulkier 1,3-diisopropyl-4,5-dimethylimidazolin-ylidene [20–22]. Stabilisation using external donors is not necessary for isolation when extremely bulky Cp′ ligands [23–25] are applied, leading to "self-stabilised"

halido-bridged dimers [{Fe($\eta^5$-Cp′)($\mu$-X)}$_2$], according to single-crystal X-ray diffraction (XRD) (Cp′ = C$_5$iPr$_5$, X = Br; Cp′ = C$_5$HiPr$_4$, X = Br, I; Cp′ = C$_5$H$_2$-1,2,4-$t$Bu$_3$, X = Br, I; Cp′ = C$_5$($p$-C$_6$H$_4$Et)$_5$, X = Br) [26–29]. Manners and Walter independently found that [{Fe($\eta^5$-C$_5$H$_2$-1,2,4-$t$Bu$_3$)($\mu$-I)}$_2$] undergoes heterolytic cleavage in toluene, affording [Fe($\eta^5$-C$_5$H$_2$-1,2,4-$t$Bu$_3$)(C$_7$H$_8$)]$^+$ and [Fe($\eta^5$-C$_5$H$_2$-1,2,4-$t$Bu$_3$)I$_2$]$^-$ [30,31]. In the same vein, deaggregation of [{Fe($\eta^5$-C$_5$H$_2$-1,2,4-$t$Bu$_3$)($\mu$-I)}$_2$] was achieved through reaction with NR$_4$I (R = Et, $n$Bu), giving rise to the formation of NR$_4$[Fe($\eta^5$-C$_5$H$_2$-1,2,4-$t$Bu$_3$)I$_2$] [30,31]. The only other closely related compound is [Fe($\eta^5$-C$_5$H$_2$-1,2,4-$t$Bu$_3$)($\mu$-Br)$_2$Na(DME)$_2$], which Sitzmann obtained through serendipity and in trace amounts only in the preparation of [{Fe($\eta^5$-C$_5$H$_2$-1,2,4-$t$Bu$_3$)($\mu$-Br)}$_2$] from [FeBr$_2$(DME)] and the corresponding sodium cyclopentadienide in DME [32]. This dinuclear complex might be viewed as a contact ion pair [Na(DME)$_2$][Fe($\eta^5$-C$_5$H$_2$-1,2,4-$t$Bu$_3$)Br$_2$], thus exhibiting, cum grano salis, the [Fe($\eta^5$-C$_5$H$_2$-1,2,4-$t$Bu$_3$)Br$_2$]$^-$ anion. In view of the mature state of half-sandwich iron(II) chemistry [1], the paucity of compounds containing simple anions of the type [Fe($\eta^5$-Cp′)X$_2$]$^-$ is quite surprising. Together with the enormous popularity of the Cp* ligand [25], this prompted us to address the synthesis of compounds containing [Fe($\eta^5$-Cp*)X$_2$]$^-$ (X = Cl, Br, I).

## 2. Results and Discussion

The synthesis of our target compounds (Scheme 1) was inspired by the work of Manners and of Walter mentioned above [30,31].

FeX$_2$ $\xrightarrow[\text{THF}]{\begin{array}{l}\text{1. MCp*, }-60\ °C\text{ to }-20\ °C\\\text{2. ER}_4\text{X, }-20\ °C\text{ to rt}\end{array}}$ [Fe(Cp*)X$_2$]ER$_4$

**Scheme 1.** Synthesis of the target compounds (X = Cl, Br, I; M = Li, K; ER$_4$ = N$n$Pr$_4$, PPh$_4$).

The addition of N$n$Pr$_4$Cl (1 equiv.) to [Fe($\eta^5$-Cp*)Cl], generated in situ from LiCp* and FeCl$_2$ in THF at low temperatures, afforded a green solution. LiCl precipitated through the addition of toluene and was subsequently removed through filtration. Storing of the filtrate at −40 °C afforded N$n$Pr$_4$[Fe($\eta^5$-Cp*)Cl$_2$] as very air-sensitive green crystals with a 60% yield. The use of N$n$Pr$_4$Br instead of N$n$Pr$_4$Cl furnished N$n$Pr$_4$[Fe($\eta^5$-Cp*)BrCl] with a 39% yield. When KCp* was used instead of LiCp*, the yields were slightly lower (by ≤8%). Both compounds were structurally characterised using XRD. Their molecular structures are shown in Figures 1 and 2, and the pertinent metric parameters are collected in Table 1. Not surprisingly, the [Fe($\eta^5$-Cp*)BrCl]$^-$ anion exhibits a disorder of the halogen atoms.

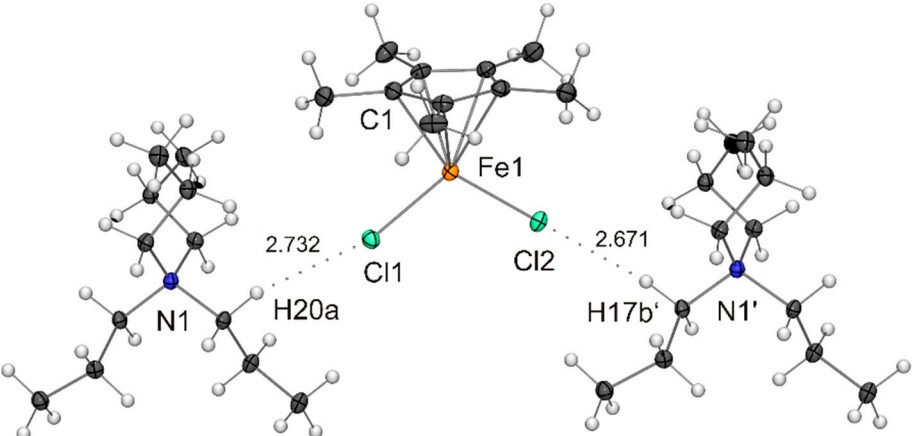

**Figure 1.** Molecular structure of N$n$Pr$_4$[Fe($\eta^5$-Cp*)Cl$_2$] in the crystal (ORTEP with 50% probability ellipsoids). The anion exhibits CH···Cl contacts compatible with weak hydrogen bonds (indicated with dotted lines) to two tetra-$n$-propylammonium cations, which are both shown.

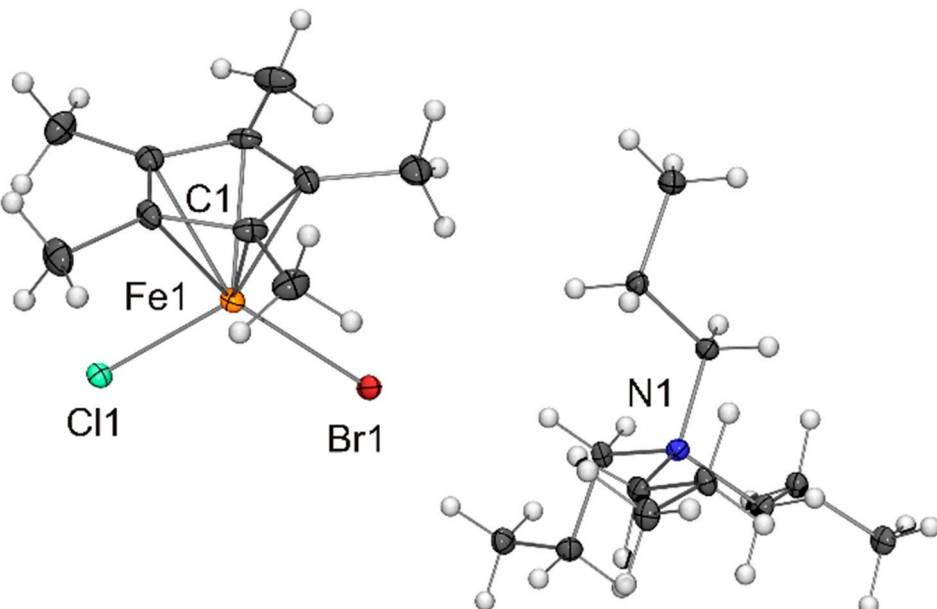

**Figure 2.** Molecular structure of NnPr$_4$[Fe($\eta^5$-Cp*)BrCl] in the crystal (ORTEP with 50% probability ellipsoids). The atom sites with the higher occupancy (58%) of the disordered halogen atoms are shown. The anion is engaged in CH···X contacts (X = Cl, Br) with neighbouring cations (not shown).

**Table 1.** Selected metric parameters (distances in Å, angles in deg) of the compounds in this study and, for comparison, of previously reported closely related compounds.

| | Fe–Cp*$_{centroid}$ | Fe–X | X–Fe–X |
|---|---|---|---|
| NnPr$_4$[Fe($\eta^5$-Cp*)Cl$_2$] | 1.975 | 2.2953(8) 2.2814(8) | 106.27(3) |
| NnPr$_4$[Fe($\eta^5$-Cp*)BrCl] [1] | 1.970 | 2.27(2) [2] 2.357(9) [3] | 109.0(6) |
| | 1.958 | 2.432(5) 2.406(5) | 103.8(2) |
| | 1.961 | 2.404(5) 2.446(5) | 103.8(2) |
| NnPr$_4$[Fe($\eta^5$-Cp*)Br$_2$] [4,5] | 1.999 | 2.415(5) 2.431(5) | 104.8(2) |
| | 1.995 | 2.405(5) 2.415(5) | 103.6(2) |
| PPh$_4$[Fe($\eta^5$-Cp*)Cl$_2$] | 1.988 | 2.288(2) 2.284(2) | 107.07(7) |
| PPh$_4$[Fe($\eta^5$-Cp*)Br$_2$] | 1.972 | 2.4278(8) | 107.86(5) |
| PPh$_4$[Fe($\eta^5$-Cp*)I$_2$] | 1.958 | 2.6201(5) | 106.56(3) |
| NnBu$_4$[Fe($\eta^5$-C$_5$H$_2$-1,2,4-tBu$_3$)I$_2$] [6] | 1.989 | 2.7003(6) 2.6144(6) | 102.20(2) |
| [Na(DME)$_2$][Fe($\eta^5$-C$_5$H$_2$-1,2,4-tBu$_3$)Br$_2$] [7] | 1.967 | 2.4633(7) 2.4316(7) | 102.36(2) |

[1] Disorder of the halogen atoms; the atom sites with the higher occupancy (58%) were chosen. [2] X = Cl. [3] X = Br.
[4] Four cations and anions are each present in the asymmetric unit. [5] Caution: the structure solution lacks quality because the arrangement and disorder of the tetra-*n*-propylammonium cations imposes non-crystallographic symmetry. [6] Ref. [31]. [7] Ref. [32].

Our attempts to prepare NnPr$_4$[Fe($\eta^5$-Cp*)Br$_2$] in an analogous way from Kölle's compound [Fe($\eta^5$-Cp*)Br] and NnPr$_4$Br furnished the product with a 33% yield but invariably afforded crystals whose structural investigation using XRD was fraught with problems due to severe cation disorder. Our best result is shown in Figure S1 in the Supporting Information. Although bond lengths and angles are given only for the heavy atoms in Table 1, these data should be treated with particular caution in the case of NnPr$_4$[Fe($\eta^5$-Cp*)Br$_2$],

where they are not taken into consideration for our discussion. The problems encountered with the tetra-*n*-propylammonium cation prompted us to use the tetraphenylphosphonium cation instead. The preparation of $PPh_4[Fe(\eta^5\text{-}Cp^*)X_2]$ (X = Cl, Br, I) through the addition of $PPh_4X$ (1 equiv.) to $[Fe(\eta^5\text{-}Cp^*)X]$ (prepared in situ from $FeX_2$ and $KCp^*$) turned out to be straightforward, although the isolated yields were unsatisfactorily poor (21% at most), probably due to the much lower solubility of $PPh_4X$ in comparison to $NnPr_4X$. A trend towards even lower yields was observed when $LiCp^*$ was used instead of $KCp^*$, which is very likely due to the concurrence of two unfavourable factors, namely the comparatively poor solubility of the tetraphenylphosphonium salts and the comparatively high solubility of the lithium salts in organic solvents of low polarity. In contrast to this, and as already noted above, $LiCp^*$ was found to be slightly more effective than $KCp^*$ in the synthesis of $NnPr_4[Fe(\eta^5\text{-}Cp^*)X_2]$. The product was obtained as crystals suitable for XRD in each case, and no disorder problems were encountered, as anticipated. The molecular structures of $PPh_4[Fe(\eta^5\text{-}Cp^*)X_2]$ are shown in Figure 3 (X = Cl), Figure 4 (X = Br) and Figure 5 (X = I).

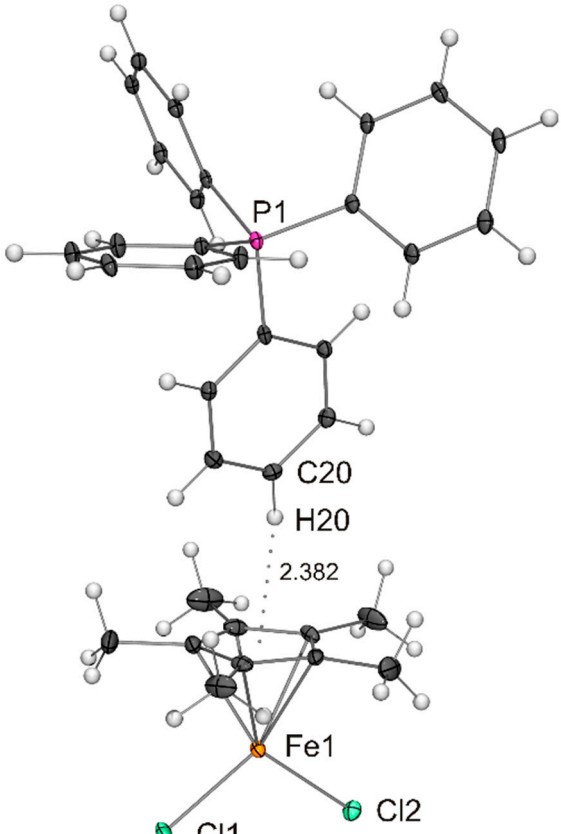

**Figure 3.** Molecular structure of $PPh_4[Fe(\eta^5\text{-}Cp^*)Cl_2]$ in thecrystal (ORTEP with 50% probability ellipsoids). The CH···π interaction between cation and anion is indicated with a dotted line. The anion is engaged in CH···Cl contacts with neighbouring cations (not shown).

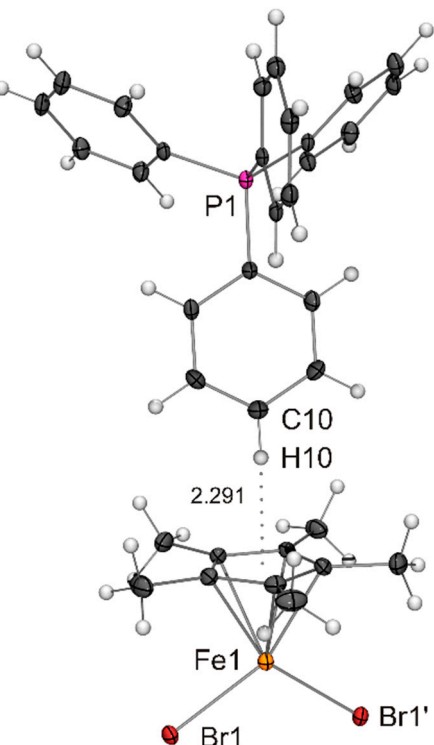

**Figure 4.** Molecular structure of $PPh_4[Fe(\eta^5\text{-}Cp^*)Br_2]$ in the crystal (ORTEP with 50% probability ellipsoids). The CH$\cdots\pi$ interaction between cation and anion is indicated with a dotted line. The anion is engaged in CH$\cdots$Br contacts with neighbouring cations (not shown).

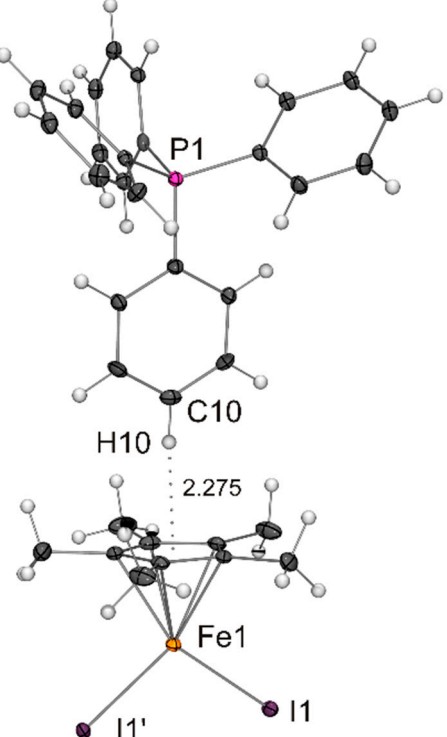

**Figure 5.** Molecular structure of $PPh_4[Fe(\eta^5\text{-}Cp^*)I_2]$ in thecrystal (ORTEP with 50% probability ellipsoids). The CH$\cdots\pi$ interaction between cation and anion is indicated with a dotted line. The anion is engaged in CH$\cdots$I contacts with neighbouring cations (not shown).

The compounds listed in Table 1 exhibit very similar iron–cyclopentadienyl ring centroid distances between 1.96 and 1.99 Å, which is much larger than the corresponding distances in the ferrocenes [Fe($\eta^5$-Cp*)$_2$] (1.65 Å) [33] and [Fe($\eta^5$-C$_5$H$_2$-1,2,4-$t$Bu$_3$)$_2$] (1.72 Å), [34] and marginally larger than those in the open-shell half-sandwich iron(II) complexes [Fe($\eta^5$-Cp*){N(SiMe$_3$)$_2$}] (1.90 Å) [35], [Fe($\eta^5$-C$_5i$Pr$_5$){N(SiMe$_3$)$_2$}] (1.92 Å) [14], and [{Fe($\eta^5$-C$_5$H$_2$-1,2,4-$t$Bu$_3$)($\mu$-X)}$_2$] (1.92 and 1.93 Å for X = Br and I, respectively) [27,28]. The differences in the Fe–X bond lengths observed for X = Cl, Br, and I are in accord with the different radii of the halogen atoms. A particularly good agreement is achieved with Pauling's tetrahedral covalent radii, which reflect a convolution of covalent and dative bonding, the values being 0.99, 1.11, and 1.28 Å for Cl, Br, and I, respectively [36]. Not surprisingly, the X–Fe–X angles of the Cp* complexes are wider (by ca. 5°) than those of the congeners containing the bulkier C$_5$H$_2$-1,2,4-$t$Bu$_3$ ligand, whose comparatively less symmetric nature may be the reason for the significant difference in the two Fe–I bond lengths ($\Delta d$ 0.09 Å) in the anion of N$n$Bu$_4$[Fe($\eta^5$-C$_5$H$_2$-1,2,4-$t$Bu$_3$)I$_2$]. The tetraalkylammonium cations are engaged in CH···X contacts compatible with weak hydrogen bonds (indicated as dotted lines in Figure 1; not shown for the disordered species in Figures 2 and 3) [37,38]. The contacts of the two halogen atoms are almost equidistant in each case (CH···Cl 2.67 and 2.73 Å for N$n$Pr$_4$[Fe($\eta^5$-Cp*)Cl$_2$], and CH···I 3.10 and 3.14 Å for N$n$Bu$_4$[Fe($\eta^5$-C$_5$H$_2$-1,2,4-$t$Bu$_3$)I$_2$]). The PPh$_4^+$ cations interact with the [Fe($\eta^5$-Cp*)X$_2$]$^-$ anions through phenyl CH···X contacts (2.76–2.95, 2.96 and 3.09–3.15 Å for X = Cl, Br and I, respectively; not shown in Figures 3–5). In addition, the *para*-H atom of a phenyl ring points towards the centre of the Cp* ligand (phenyl CH···C 2.53–2.74 Å, phenyl CH···Cp* ring centroid 2.28–2.38 Å, shown as dotted lines in Figures 3–5), indicating a CH···$\pi$ interaction [39,40] similar to that in the T-shaped benzene dimer [41–46] for which a CH···C$_6$H$_6$ ring centroid distance of 2.25 Å was computed recently [47].

The electronic structure of the anion of N$n$Bu$_4$[Fe($\eta^5$-C$_5$H$_2$-1,2,4-$t$Bu$_3$)I$_2$] was scrutinised using SQUID magnetometry, EPR spectroscopy, and ab initio Complete Active Space Self Consistent Field-Spin Orbit calculations, which revealed a high-spin d$^6$ iron(II) centre with a strongly anisotropic $S$ = 2 ground state [31]. This in-depth study by Manners makes an analogous investigation of our closely related compounds dispensable. The paramagnetic nature of their [Fe($\eta^5$-Cp*)X$_2$]$^-$ anions is clearly evident from the NMR spectra. The Cp* ligand gives rise to a $^1$H NMR signal at $\delta \approx 200$ ppm. This may be compared with the data reported for the substituted cyclopentadienyl ligands of [{Fe($\eta^5$-C$_5i$Pr$_5$)($\mu$-Br)}$_2$] in C$_6$D$_6$ [$\delta(^1$H) = 95.7 (C$H$Me$_2$), 11.3 (CH$Me_2$), and −117.3 ppm (CH$Me_2$)] [26] and of N$n$Bu$_4$[Fe($\eta^5$-C$_5$H$_2$-1,2,4-$t$Bu$_3$)I$_2$] in THF-$d_8$ [$\delta(^1$H) = −20.3 and −31.4 ppm (2 × $t$Bu)] [31].

Kölle demonstrated the successful generation of the highly reactive compound [Fe($\eta^5$-Cp*)Br] through a trapping reaction with carbon monoxide at −80 °C, which furnished the diamagnetic carbonyl complex [Fe($\eta^5$-Cp*)Br(CO)$_2$] in a 59% yield [2]. In the same vein, Walter obtained [Fe($\eta^5$-C$_5$H$_2$-1,2,4-$t$Bu$_3$)I(CO)$_2$] through carbonylation of the "self-stabilised" halido-bridged dimer [{Fe($\eta^5$-C$_5$H$_2$-1,2,4-$t$Bu$_3$)($\mu$-I)}$_2$] with CO at room temperature in an 80% yield [28]. We have studied the carbonylation of our target compounds exemplarily with N$n$Pr$_4$[Fe($\eta^5$-Cp*)Cl$_2$] and observed an essentially quantitative reaction with CO under the same mild conditions, affording the well-known carbonyl complex [Fe($\eta^5$-Cp*)Cl(CO)$_2$] [48,49]. The crystal structure of this compound reported in 1988 was determined at room temperature [49], which prompted us to redetermine the structure at 100 K (see the Supporting Information).

## 3. Materials and Methods

**Experimental Details.** All reactions were performed in an inert atmosphere (argon or dinitrogen) using standard Schlenk techniques or a conventional glovebox. Solvents were dried with a commercial Solvent Purification System (M. Braun, Garching, Germany, MB SPS 7), degassed and stored over 3 Å molecular sieves under inert atmosphere. Starting materials were procured from standard commercial sources and used as received. LiCp* and KCp* were synthesised through deprotonation of pentamethylcyclopentadiene in

*n*-hexane with *n*-butyllithium and potassium metal, respectively, and isolated through filtration or centrifugation. NMR spectra were recorded with Varian MR-400 and Varian NMRS-500 spectrometers operating at 400 and 500 MHz, respectively, for [1]H. Elemental analyses were carried out with a HEKAtech Euro EA-CHNS elemental analyser at the Institute of Chemistry, University of Kassel, Germany.

**N*n*Pr$_4$[Fe(η$^5$-Cp*)Cl$_2$]:** A Schlenk tube charged with LiCp* (176 mg, 1.24 mmol) and FeCl$_2$ (156 mg, 1.23 mmol) was cooled to −60 °C. THF (3 mL) cooled to the same temperature was added. The stirred mixture was allowed to warm up to −20 °C. N*n*Pr$_4$Cl (275 mg, 1.24 mmol) was added. The stirred mixture was allowed to warm up to ambient temperature and was subsequently filtered through a Celite pad. Toluene (ca. 3 mL) was slowly added to the green filtrate until formation of an essentially colourless precipitate was observed. Insoluble material was removed via filtration through a Celite pad. Storing of the filtrate at −40 °C afforded the product as green crystals, which were separated from the yellow mother liquor, washed with *n*-hexane (5 mL), and dried under vacuum. Yield, 327 mg (60%). Elemental analysis for C$_{22}$H$_{43}$NCl$_2$Fe (448.34 g/mol): calculated (%): C 58.94, H 9.67, N 3.12. Found (%): 58.18, H 9.37, N 3.21. [1]H NMR (400 MHz, THF-$d_8$): $\delta$ 194.4 (15H, s, $\nu_{\frac{1}{2}}$ = 380 Hz, Cp*), 16.8 (8H, s, $\nu_{\frac{1}{2}}$ = 270 Hz, (C*H$_2$*)$_2$CH$_3$), 9.9 (8H, s, $\nu_{\frac{1}{2}}$ = 220 Hz, (C*H$_2$*)$_2$CH$_3$), 1.3 (12H, s, $\nu_{\frac{1}{2}}$ = 270 Hz, (CH$_2$)$_2$C*H$_3$*).

**N*n*Pr$_4$[Fe(η$^5$-Cp*)BrCl]:** This compound was obtained through a procedure analogous to that described above for N*n*Pr$_4$[Fe(η$^5$-Cp*)Cl$_2$] using LiCp* (130 mg, 0.91 mmol), FeCl$_2$ (116 mg, 0.92 mmol), and N*n*Pr$_4$Br (245 mg, 0.92 mmol) in THF (3 mL). Yield, 175 mg (39%). Elemental analysis for C$_{22}$H$_{43}$NBrClFe (492.79 g/mol): calculated (%): C 53.62, H 8.80, N 2.84. Found (%): C 54.24, H 8.81, N 2.48. [1]H NMR (400 MHz, THF-$d_8$): $\delta$ 206.5 (15H, s, $\nu_{\frac{1}{2}}$ = 2860 Hz, Cp*), 25.8 (8H, s, $\nu_{\frac{1}{2}}$ = 500 Hz, (C*H$_2$*)$_2$CH$_3$), 15.1 (8H, s, $\nu_{\frac{1}{2}}$ = 350 Hz, (C*H$_2$*)$_2$CH$_3$), 2.7 (12H, s, $\nu_{\frac{1}{2}}$ = 650 Hz, (CH$_2$)$_2$C*H$_3$*).

**N*n*Pr$_4$[Fe(η$^5$-Cp*)Br$_2$]:** This compound was obtained through a procedure analogous to that described above for N*n*Pr$_4$[Fe(η$^5$-Cp*)Cl$_2$] using LiCp* (65 mg, 0.46 mmol), FeBr$_2$ (99 mg, 0.46 mmol) and N*n*Pr$_4$Br (122 mg, 0.46 mmol) in THF (1.5 mL). Yield, 81 mg (33%). An analytical sample was obtained through recrystallization from benzene. Elemental analysis for C$_{22}$H$_{43}$NBr$_2$Fe·$\frac{1}{2}$C$_6$H$_6$ (576.29 g/mol): calculated (%): C 52.10, H 8.05, N 2.43. Found (%): C 52.18, H 8.24, N 1.76. [1]H NMR (400 MHz, THF-$d_8$): $\delta$ 203.5 (15H, s, $\nu_{\frac{1}{2}}$ = 560 Hz, Cp*), 16.7 (8H, s, $\nu_{\frac{1}{2}}$ = 310 Hz, (C*H$_2$*)$_2$CH$_3$), 10.7 (8H, s, $\nu_{\frac{1}{2}}$ = 240 Hz, (C*H$_2$*)$_2$CH$_3$), 2.23 (12H, s, $\nu_{\frac{1}{2}}$ = 190 Hz, (CH$_2$)$_2$C*H$_3$*).

**PPh$_4$[Fe(η$^5$-Cp*)Cl$_2$]:** A Schlenk tube charged with KCp* (40 mg, 0.23 mmol) and FeCl$_2$ (29 mg, 0.23 mmol) was cooled to −60 °C. THF (0.5 mL) cooled to the same temperature was added. The stirred mixture was allowed to warm up to −20 °C. PPh$_4$Cl (86 mg, 0.23 mmol) was added. The stirred mixture was allowed to warm up to ambient temperature and was subsequently filtered through a Celite pad. The yellow filtrate was carefully layered with *n*-hexane, resulting in the slow formation of yellow crystals, which were separated from the mother liquor, washed with *n*-hexane (2 mL), and dried under vacuum. Yield, 8 mg (6%). In view of the unsatisfactorily low yield, elemental analysis was not performed for this compound. [1]H NMR (500 MHz, THF-$d_8$): $\delta$ 188.2 (15H, s, $\nu_{\frac{1}{2}}$ = 310 Hz, Cp*), 13.2 (8H, $\nu_{\frac{1}{2}}$ = 100 Hz, Ph), 10.7 (8H, $\nu_{\frac{1}{2}}$ = 100 Hz, Ph), 10.2 (4H, $\nu_{\frac{1}{2}}$ = 80 Hz, Ph).

**PPh$_4$[Fe(η$^5$-Cp*)Br$_2$]:** This compound was obtained through a procedure analogous to that described above for PPh$_4$[Fe(η$^5$-Cp*)Cl$_2$] using KCp* (40 mg, 0.23 mmol), FeBr$_2$ (50 mg, 0.23 mmol), and PPh$_4$Br (96 mg, 0.23 mmol) in THF (0.5 mL). Yield, 12 mg (8%). In view of the unsatisfactorily low yield, elemental analysis was not performed for this compound. [1]H NMR (500 MHz, THF-$d_8$): $\delta$ 193.7 (15H, s, $\nu_{\frac{1}{2}}$ = 650 Hz, Cp*), 11.3 (8H, $\nu_{\frac{1}{2}}$ = 160 Hz, Ph), 9.2 (8H, $\nu_{\frac{1}{2}}$ = 190 Hz, Ph), 8.6 (4H, $\nu_{\frac{1}{2}}$ = 190 Hz, Ph).

**PPh$_4$[Fe(η$^5$-Cp*)I$_2$]:** This compound was obtained through a procedure analogous to that described above for PPh$_4$[Fe(η$^5$-Cp*)Cl$_2$] using KCp* (40 mg, 0.23 mmol), FeI$_2$ (71 mg,

0.23 mmol), and PPh$_4$I (107 mg, 0.23 mmol) in THF (0.5 mL). Yield, 38 mg (21%). In view of the unsatisfactorily low yield, elemental analysis was not performed for this compound. $^1$H NMR (500 MHz, THF-$d_8$): δ 209.9 (15H, s, ν$_{\frac{1}{2}}$ = 530 Hz, Cp*), 10.1 (8H, ν$_{\frac{1}{2}}$ = 60 Hz, Ph), 9.1 (8H, ν$_{\frac{1}{2}}$ = 60 Hz, Ph), 8.7 (4H, ν$_{\frac{1}{2}}$ = 60 Hz, Ph).

**[Fe(η$^5$-Cp*)Cl(CO)$_2$]:** A solution of N*n*Pr$_4$[Fe(η$^5$-Cp*)Cl$_2$] (40 mg, 0.09 mmol) in THF (2 mL) was subjected to an atmospheric pressure of CO, which led to an immediate colour change from green to red. The solution was stirred for 10 min. Volatile components were removed under vacuum. Benzene (0.7 mL) was added to the residue. Insoluble material was removed via filtration through a Celite pad. Slow evaporation of the filtrate afforded the product as red crystals. Yield, 23 mg (92%). Spectroscopic data were found to be in good agreement with published values [48,49].

**X-ray Crystallography:** For all data collections, a single crystal was mounted on a micro-mount, and all geometric and intensity data were taken from this sample through ω-scans at 100(2) K. Data collections were carried out either on a Stoe StadiVari diffractometer equipped with a 4-circle goniometer and a DECTRIS Pilatus 200K detector (for N*n*Pr$_4$[Fe(η$^5$-Cp*)Cl$_2$], N*n*Pr$_4$[Fe(η$^5$-Cp*)Br$_2$], and PPh$_4$[Fe(η$^5$-Cp*)Cl$_2$]) or on a Stoe IPDS2 diffractometer equipped with a 2-circle goniometer and an area detector (for N*n*Pr$_4$[Fe(η$^5$-Cp*)BrCl], PPh$_4$[Fe(η$^5$-Cp*)Br$_2$], PPh$_4$[Fe(η$^5$-Cp*)I$_2$], and [Fe(η$^5$-Cp*)Cl(CO)$_2$]). The data sets were corrected for absorption (through multi scan), Lorentz, and polarisation effects. The structures were solved using direct methods (SHELXT 2014/7) [50] and refined using alternating cycles of least-squares refinements against $F^2$ (SHELXL2014/7) [50]. H atoms were included in the models in calculated positions with the 1.2-fold isotropic displacement parameter of their bonding partner. Experimental details for each diffraction experiment are given in Table S1 (Supplementary Materials). CCDC 2300615–2300621 contain supplementary crystallographic data for this paper. These data can be obtained free of charge from The Cambridge Crystallographic Data Centre, www.ccdc.cam.uk/structures (accessed on 11 October 2023).

## 4. Conclusions

Thermally stable half-sandwich iron(II) dihalido complexes of the type [Fe(η$^5$-Cp′)X$_2$]$^-$ reported in the literature have so far been limited to a small number of salts containing the anion [Fe(η$^5$-C$_5$H$_2$-1,2,4-*t*Bu$_3$)I$_2$]$^-$. We extended this to homologues [Fe(η$^5$-Cp*)X$_2$]$^-$ (X = Cl, Br, I) containing the widely used Cp* ligand. Corresponding ionic compounds ER$_4$[Fe(η$^5$-Cp*)X$_2$] are easily accessible from FeX$_2$, MCp* (M = Li, K) and a suitable halide source R$_4$EX (E = N, P). Not surprisingly, they are very air-sensitive not only in solution but also in the solid state and consequently should be handled under rigorously inert conditions. While yields of up to 60% could be achieved with ER$_4$ = N*n*Pr$_4$, unsatisfactorily low yields were obtained with ER$_4$ = PPh$_4$, which, however, turned out to be superior to N*n*Pr$_4$ in terms of the quality of crystals needed for XRD. The high-yield synthesis of [Fe(η$^5$-Cp*)Cl(CO)$_2$] from N*n*Pr$_4$[Fe(η$^5$-Cp*)Cl$_2$] and CO under mild conditions exemplarily demonstrates that such anions are amenable to halido ligand substitution reactions and may thus provide facile access to a range of pentamethylcyclopentadienyliron half-sandwich complexes.

**Supplementary Materials:** The following supporting information can be downloaded at https://www.mdpi.com/article/10.3390/inorganics11110437/s1: Table S1: X-ray crystallographic details; Figure S1: Molecular structure of N*n*Pr$_4$[Fe(η$^5$-Cp*)Br$_2$] in the crystal; Figure S2: $^1$H NMR spectrum of N*n*Pr$_4$[Fe(η$^5$-Cp*)Cl$_2$]; Figure S3: $^1$H NMR spectrum of N*n*Pr$_4$[Fe(η$^5$-Cp*)BrCl]; Figure S4: $^1$H NMR spectrum of N*n*Pr$_4$[Fe(η$^5$-Cp*)Br$_2$]; Figure S5: $^1$H NMR spectrum of PPh$_4$[Fe(η$^5$-Cp*)Cl$_2$]; Figure S6: $^1$H NMR spectrum of PPh$_4$[Fe(η$^5$-Cp*)Br$_2$]; Figure S7: $^1$H NMR spectrum of PPh$_4$[Fe(η$^5$-Cp*)I$_2$]; Figure S8: $^1$H NMR spectrum of [Fe(η$^5$-Cp*)Cl(CO)$_2$]; Figure S9: $^{13}$C NMR spectrum of [Fe(η$^5$-Cp*)Cl(CO)$_2$].

**Author Contributions:** Conceptualisation, U.S.; formal analysis, C.B.; investigation, J.Z. and C.B.; methodology, J.Z.; project administration, U.S.; supervision, U.S.; writing original draft, U.S.; writing review and editing, U.S. All authors have read and agreed to the published version of the manuscript.

**Funding:** This research received no external funding.

**Data Availability Statement:** The data presented in this study are available in the supporting information.

**Conflicts of Interest:** The authors declare no conflict of interest.

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
