# Peer review of "Ammonium and Phosphonium Salts Containing Monoanionic Iron(II) Half-Sandwich Complexes [Fe(η5-Cp*)X2] (X = Cl − I)"

_inorganics, doi:10.3390/inorganics11110437_

Round 1

Reviewer 1 Report

Comments and Suggestions for Authors

This is a nice piece of research dealing with the syntheses of a series of ionic complexes of composition {[FeCp*X2]+[ER4]-} Despite the structural simplicity of these compounds, there are not many examples of similar, coordinatively unsaturated 16-electron derivatives of the FeCp half-sandwich moiety that are stable enough to be isolated and characterized.

These compounds (particularly the N(nPr)4 ones) could be helpful as organometallic precursors, and I only missed some comments on the thermal stability of these compounds and sensitivity to the atmospheric agents, which might prove very helpful to those who wish to apply this chemistry to their research.

In general, the article is clear and very well written. Just a couple of minor issues: I) It would be better to write N(nPr)4 rather than NnPr4 (the Nn might be taken for a generic atomic symbol). ii) The sentence: "In contrast to the synthesis of NnPr4[Fe(5‐Cp*)X2], a trend towards even lower yields was observed when LiCp* was used instead of KCp*" is somewhat confusing because it is not mentioned whether the reaction with LiCp* was systematically attempted, and the yields were even lower than with KCp*. Since the yields with KCp* were low enough, LiCp* must be essentially ineffective. This is likely a solubility issue due to the concurrence of two unfavorable factors: the insolubility of the tetraphenylphosphonium reagent and the higher solubility of lithium salts in nonpolar solvents.

Author Response

We have uploaded our response to the editor and reviewers as a single PDF file.

Reviewer 2 Report

Comments and Suggestions for Authors

The manuscript is a nice little piece of work describing stabilization of instable Cp*FeX species by quaternary amonium and phosphonium halide salts. The formed species ER4[Cp*FeX2] was characterized by NMR, X-ray and in some cases by elemental analysis. Author demonstrated similar behavior of ER4[Cp*FeX2] and (in situ generated) Cp*FeX on just one reaction (carbonylation).

I would expect little bit deeper characterization of the species (eg. IR and ESI-MS, melting points). Also utilization of TGA for quantification of thermal stability is recommended. In addition, I strongly recommend to extend the investigation of reactivity of ER4[Cp*FeX2] (eg. towards preparation of ferrocenes with different ring).

However, I supported publication of the manuscript in a present form

minor comment:

-half-width of  broad signals in 1H NMR should be rounded to tens (uncertainty of assignment)

Author Response

(The authors gave the same response as above.)
